# An Edge Device Framework in SEMAR IoT Application Server Platform

Yohanes Yohanie Fridelin Panduman [1], Nobuo Funabiki [1,*], Sho Ito [1], Radhiatul Husna [1], Minoru Kuribayashi [1], Mitsuhiro Okayasu [1], Junya Shimazu [1] and Sritrusta Sukaridhoto [2]

[1] Graduate School of Natural Science and Technology, Okayama University, Okayama 700-8530, Japan; p8f01q6f@s.okayama-u.ac.jp (Y.Y.F.P.); p3ti3fqh@s.okayama-u.ac.jp (S.I.); pwmn7i7q@s.okayama-u.ac.jp (R.H.); kminoru@okayama-u.ac.jp (M.K.); mitsuhiro.okayasu@utoronto.ca (M.O.); p5835ic4@s.okayama-u.ac.jp (J.S.)

[2] Department of Informatic and Computer, Politeknik Elektronika Negeri Surabaya, Surabaya 60111, Indonesia; dhoto@pens.ac.id

\* Correspondence: funabiki@okayama-u.ac.jp

**Abstract:** Nowadays, the *Internet of Things (IoT)* has become widely used at various places and for various applications. To facilitate this trend, we have developed the IoT application server platform called *SEMAR (Smart Environmental Monitoring and Analytical in Real-Time)*, which offers standard features for collecting, displaying, and analyzing sensor data. An *edge device* is usually installed to connect sensors with the server, where the interface configuration, the data processing, the communication protocol, and the transmission interval need to be defined by the user. In this paper, we proposed an *edge device framework* for *SEMAR* to remotely optimize the edge device utilization with three phases. In the *initialization phase*, it automatically downloads the configuration file to the device through *HTTP* communications. In the *service phase*, it converts data from various sensors into the standard data format and sends it to the server periodically. In the *update phase*, it remotely updates the configuration through *MQTT* communications. For evaluations, we applied the proposal to the *fingerprint-based indoor localization system (FILS15.4)* and the *data logging system*. The results confirm the effectiveness in utilizing *SEMAR* to develop IoT application systems.

**Keywords:** Internet of Things; edge device; framework; application server platform; SEMAR





## 1. Introduction

Currently, the *Internet of Things (IoT)* is receiving much attention from both industries and academics as an emerging technology that uses the Internet infrastructure to connect physical worlds to cyberspaces [1]. The IoT application infrastructure is continuously being extended to become more ubiquitous around the world and is composed of numerous physical devices distributed across multiple domains [2]. In this context, the success of an IoT application system depends on the ability to collect, manage, and analyze the data easily and flexibly, as well as to distribute it to users and other systems efficiently [3,4]. Nowadays, the amount of data generated by sensor devices is increasing rapidly with the availability of diverse network connectivity and various protocol services; IoT application system developers should design and build these systems considering standardizations with heterogeneous device management.

In an IoT application system, *edge computing* is often adopted to bring computing capabilities for data processing to locations closer to sensors or target devices [5]. Some IoT applications may require low latency and real-time data processing, which cloud servers cannot provide [6,7]. Due to the diversity of sensor resources, the introduction of *edge computing devices* has become a valuable solution to reducing the computational complexity of data processing in cloud servers [8]. Edge computing devices enable various functions at the edges of networks before sending data to the server and can increase the efficiency of data processing [9]. It also offers the data conversion capability to convert raw data

to the standard data format. It is expected that the *edge device framework* was introduced to facilitate application developments in edge computing devices [10]. The framework interacts with devices in the physical world that may change over time [11]. Therefore, it should support the dynamic development of edge systems.

Recently, cloud-based solutions have been widely used for IoT application systems [12]. Instead of focusing on the implementation details, the prepared tools allow developers to focus on the implementation of logic by using functions that efficiently support the design and implementation of IoT applications [13]. However, most of the existing cloud-based solutions did not support effective and efficient developments at the edge devices level, and their technologies have often limited the interoperability with third parties.

Previously, we designed and implemented the IoT application server platform as a cloud-based solution for integrating various IoT application systems, called *SEMAR (Smart Environmental Monitoring and Analytical in Real-Time)* [14]. *SEMAR* provides standard features for collecting, displaying, processing, and analyzing sensor data from different domains. It offers *built-in* functions for data synchronizations, aggregations, and classifications with machine learning in *Big Data* environments, and *plug-in* functions for allowing other systems to access the data through the *Representational State Transfer Application Programming Interface (REST API)*.

Unfortunately, the current implementation of *SEMAR* does not facilitate deployments and implementations of *edge devices* within the context of IoT ecosystem application deployments. As an effective IoT application server platform, *SEMAR* should be able to control and manage various IoT devices remotely. It must be capable of reconfiguring IoT devices to improve their performance and utilization.

In this paper, we proposed an *edge device framework* and its implementation for *SEMAR* to facilitate the development of edge devices for IoT applications. As a popular edge device, the *Raspberry Pi* was selected for this implementation, and the image was created in the *SEMAR* server. This framework can remotely optimize the utilization of this edge device by configuring the connectivity of sensor interfaces, a data conversion approach, a data model, transmitted data, local data storage, local visualization, and the data transmission interval on the server. Actually, it provides features for downloading configuration files to the devices using *HTTP* communications, converting data from diverse sensor resources into standard data formats before delivering them to *SEMAR*, processing data using rules and filter functions, offering multiple output components for utilizing the acquired data, and enabling remote configuration updates using *Message Queue Telemetry Transport (MQTT)* services [15].

For evaluations of the proposal, we applied the edge device framework to the *fingerprint-based indoor localization system (FILS15.4)* [16,17] and the *data logging system*. These integrated systems were deployed in #1 and #2 Engineering Buildings at Okayama University, Japan. In addition, we evaluated the effectiveness of the edge device framework by investigating its computing performance and comparing it with similar research works. The results confirm the feasibility of utilizing the edge device framework in developing IoT application systems with *SEMAR*.

The rest of this paper is organized as follows: Section 2 presents related works. Section 3 describes the IoT application system architecture. Section 4 briefly reviews our previous works on *SEMAR*. Section 5 presents the design and implementation of the edge device framework. Sections 6 and 7 briefly describe the implementation in two IoT application systems. Section 8 presents comprehensive performance evaluations and a comparative analysis with similar related work. Finally, Section 9 concludes this paper with future works.

## 2. Related Works

In [18], Mahmood et al. presented a simulation of an edge computing implementation for resource allocation in IoT applications for smart cities. The result shows the effectiveness

of the edge computing layer in reducing the energy and computational resources for IoT networks.

In [19], Sarangi et al. proposed IoT applications for digital farming by using a microcontroller that connects soil moisture sensors with the mobile system as edge gateways. The edge device captures and transmits sensor data to the mobile system through Wi-Fi communications. Then, the mobile system processes the data and sends them to the cloud server. This approach presented the utilization of the mobile system for collecting and processing information at edges to reduce computational processes at the cloud level.

In [20], Oueida et al. proposed an integration of the edge computing device and the cloud service in the smart healthcare system. Edge computing was used to gather information from smart devices, process it to obtain the necessary data, and transmit it to the cloud server. The proposed system was suitable for emergency departments and other types of queuing systems.

In [21], Mach et al. proposed the concept of the mobile edge computing, which enables IoT applications to perform massive data processing at the device level. However, developers should consider three key aspects, namely, the computation decision, the resource allocation for computational processes, and the mobility management. This approach can reduce the latency of the network in IoT application systems.

In [22], Yousafzai et al. introduced a light-effect migration-based paradigm for managing computational offloading in edge networks in mobile edge computing. They investigated the impacts of edge networks on IoT applications. The evaluation results showed that the execution time for data processing and the amount of transmitted data should be considered to optimize the utilization of edge devices.

In [13], Berta et al. proposed a general end-to-end IoT platform that is composed of the cloud-based service for managing sensor data and devices of IoT applications called *Measurify*, and the tool for facilitating the construction of edge devices called *Edgine*. *Edgine* requests the local configuration and executable scripts. Then, it collects data from the sensors, processes them using downloadable scripts and stores it in the cloud. The proposed system has been installed and used for several IoT application systems. The results demonstrated the efficiency of the system by enabling developers to focus on application requirements and design decisions to define the edge system rather than on implementations.

In [23], Yang et al. proposed an edge computing framework suitable for IoT device development. This framework provides functions to configure the module hardware security, the data conversion, control, and communication to the server. It also offers advanced data processing capabilities at the edge computing level, including rule engines, data analysis, and application integration. By accessing the cloud service, this framework allows users to update the configuration through *MQTT* communications. This approach is similar to our method for updating the configuration remotely.

In [24], Kim et al. proposed plug-and-play in IoT platforms, using a web page to manage IoT devices. They utilized *Arduino* boards as edge devices that were connected to the sensors and actuators. The proposed system allows configuring the device for data collection or control actions by accessing the platform website. The implementation results indicate that the system was able to reduce the deployment complexity and increase the IoT environment dynamicity. However, they only considered the device layer and did not address the data visualization and analysis at the cloud level.

In [25], Iera et al. introduced the *Social Internet of Things (SIoT)* architecture paradigm. This architecture comprises IoT applications in objects that are registered on a social networking platform, where each object collaborates and interacts with other objects to provide specialized services. The architecture includes three elements: objects, gateway, and an *SIoT* server. Each component may consist of three layers: sensing, network, and application. It enables IoT objects to conduct high-computational processes, in contrast to only the server performing these tasks. As a common IoT architecture, the network layer is only used to connect the server and the objects. However, this architecture allows

the integration between IoT objects and provides interfaces for IoT objects and humans through network layers. Thus, it provides the development of IoT applications that interact with one another. This architecture can be considered a reference with which to improve the design of the IoT application system architecture proposed in this paper.

In [26], Cauteruccio et al. proposed the *Multi-Internet of Things (MIoT)* architecture to improve object communication in the *SIoT* architecture. In the *SIoT* architecture, IoT objects connect and collaborate with one another. It makes the complexity of data transfer increase. Thus, MIoT architecture solves this issue by considering data-driven and semantics-based aspects of data exchange between objects. Unfortunately, the proposed communication model is not suitable for dynamic IoT application scenarios, where IoT devices are dynamically added and removed.

## 3. Design of The IoT Application System Architecture

### 3.1. System Overview

In this section, we describe the design of the IoT application system architecture for generalization. Currently, there are many IoT architecture references that can be considered for developing IoT application systems. However, each IoT application system has unique designs and requirements. The common IoT application system architecture consists of three layers. The *perception layer* represents the physical devices for sensing and actuating that interact with the environment. The *network layer* represents the transport layer for data communications between layers. The *application layer* represents the application software to offer specific services for data processing [27]. There are many IoT application system architectures that need to be addressed to enhance the development of IoT applications and platforms.

In [28], Lombardi et al. presented commonly used IoT architectures such as *cloud-based* architecture, *edge-computing-based* architecture, and *Social Internet of Things (SIoT)* architecture. *Cloud-based* architecture utilizes services deployed on a cloud server to generate, process, and visualize large amounts of data for users. This architecture allows users and other services to access data at any time. *Edge-computing-based* architecture offers computational services close to the device layer by offering data processing, storage, and control capabilities. It is frequently used for industrial devices and IoT application systems that demand a quick response as a result of data processing.

In *SIoT* architecture, IoT applications are comprised of objects registered on a social networking platform, where each object collaborates and interacts with other objects to provide specific services [25]. This architecture enables IoT objects to conduct high-computational processes, as opposed to only the server performing these tasks. It enables the development of IoT applications that interact with one another. In addition, the *MIoT* architecture has been added to the *SIoT* architecture. In order to reduce the complexity of the *SIoT* architecture system, the *MIoT* architecture considers data-driven and semantics-based aspects for data exchange between objects [26].

In this paper, the concept of the IoT application system architecture was based on these references. Figure 1 illustrates the proposed architecture. It is composed of the *sensors and actuators*, *edge*, and *cloud* layers.

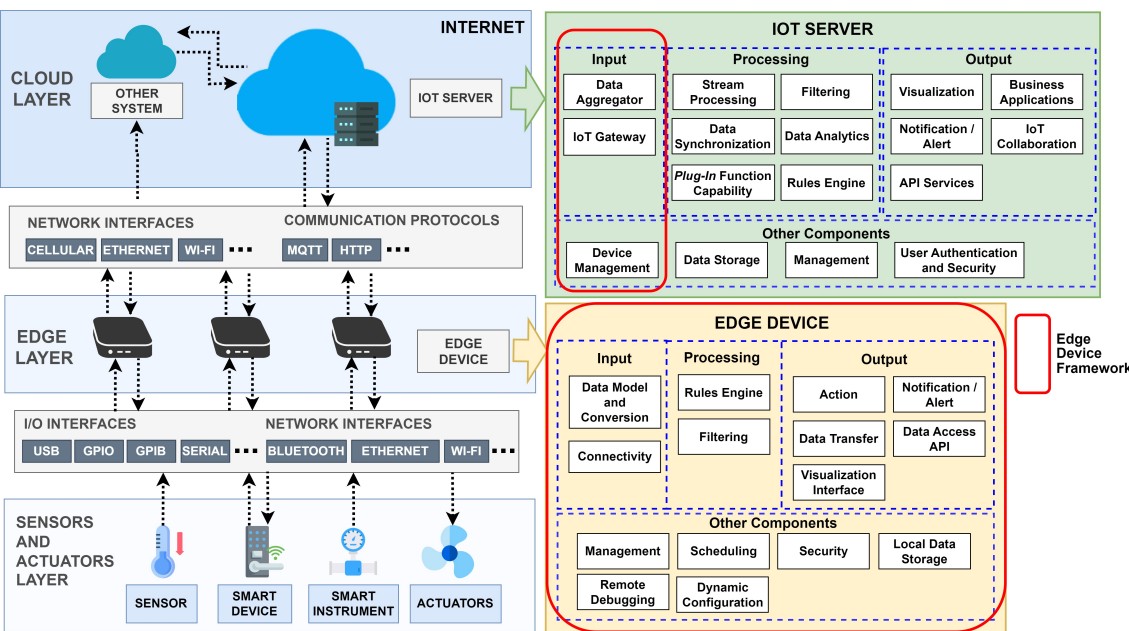

**Figure 1.** Design overview of general IoT application system architecture.

### 3.2. Sensor and Actuators Layer

In the context of the IoT application system, perception devices as IoT objects are sensors and actuators connected to a controller. Sensors are primarily used to monitor the environment by converting physical parameters into measurable electrical quantities (often voltage), while actuators provide physical actions when presented with an electrical quantity. However, with the rapid development of technologies, Internet-connected devices have become common and diverse in their application purposes.

For instance, in smart homes, developers have often utilized smart devices to improve living experiences and reduce energy consumption. These smart devices are controlled by smartphones and are integrated with cloud services through wireless networks.

The *Industrial Internet of Things (IIoT)* has been presented to connect IoT technologies to industrial machines or instruments to analyze the obtained data and optimize existing industrial processes [29]. It uses smart instrument devices for automatic data collection to enhance the condition monitoring of industrial instruments. Recently, industrial devices in the market have contained features to enable Internet-based data access to central operation management systems through Ethernet and wireless technology. In this paper, we considered smart devices and smart instruments as components in the sensor and actuator layers of the proposed architecture.

### 3.3. Edge Layer

The *edge* layer addresses the issue of the growing data volume in an IoT application system by utilizing computing capabilities of edge devices. In this section, we explain the components of the edge device—*input*, *processing*, *output*, and *other components*.

#### 3.3.1. Input Components

*Input* components should consider the connectivity of IoT devices and the method for collecting valuable data from them. The connectivity component refers to the *input/output (I/O)* and the network interfaces of the IoT device for data communications. Currently, a single-board computer, such as *Raspberry Pi*, has enabled various interfaces to accept data from a variety of devices. Among them, *General Purpose Input Output (GPIO)* is the standard interface for receiving and sending commands to/from IoT sensors and actuators. *General Purpose Interface Bus (GPIB)* is the I/O interface included in the IEEE-488 standard

for industrial instrumentation data. While *GPIO* only transmits data in signals, *GPIB* is able to handle both text data and numeric expressions.

In the context of IoT data communications, serial communication protocols are often used to transfer data among IoT devices. Each device may support different serial interfaces based on its hardware specifications. These include the *RS-232 protocol*, *Universal Serial Bus (USB)*, the *Serial Peripheral Interface (SPI)*, the *Universal Asynchronous Receiver Transmitter (UART)*, and the *Inter-Integrated Circuit (I2C)*. It is necessary to build an edge system that is able to handle different interfaces.

Various network interfaces, including Bluetooth, Ethernet, and Wi-Fi, have been introduced to connect IoT objects and edge computing devices. *Bluetooth* is widely applicable in smart devices due to its capability of low-power communications. *Ethernet* provides stability and security by wired connectivity. However, it is difficult to communicate over long distances. In IoT application systems, *IEEE 802.11 wireless LAN (Wi-Fi)* is the most popular network interface used by current smart devices and smart instruments.

Sensor devices usually generate data in different and non-standard formats. It is challenging to enable the interoperability among sensors from different companies that have different communication technologies. Therefore, the edge device requires the data conversion component to generate data in the standard format from various sensor devices. This component represents the translation process of sensor data. It requires a data model to define the valuable data structures of sensor data that are used for further processing in the edge system. *JavaScript Object Notation (JSON)* format data are frequently used for this purpose.

### 3.3.2. Processing Components

As an extension of cloud services, edge computing has similar characteristics to cloud computing. Edge computing is able to perform local data processing with minimal computational resources. Processing components in the edge layer are designed to optimize data collections and enable immediate analysis and decision-making. The filtering and the rules engine are included in these components. The *filtering* component reduces data noise and inaccuracies by applying digital filters to sensor data. Several sensors, such as the accelerometer and the gyroscope, may produce noisy data. It is necessary to reduce noises before transmitting data to a cloud server.

The *rules engine* component makes data-driven decisions in real-time. It applies various output services when rule patterns are matched. They include delivering notification messages to users and issuing action commands to actuators. The rules contain basic operations in the format of "if the specific conditions are fulfilled, then trigger the specific actions" or defined as *IF-THIS-THEN-THAT* form—for example, in an IoT application system for smart homes, "if the temperature is higher than 30 °C, then turn on the air conditioner". The rules engine in the edge layer can reduce the time required to generate the response action, compared to waiting for the server response. However, it should avoid complex rule models due to the limited computational resources of edge devices.

### 3.3.3. Output Components

The output components concern the ability of edge devices to utilize the collected data and transmit it to the cloud server or other systems. Several output components, such as the visualization interface, notification/alert, data transfer, trigger action operations, and data access API, should be considered for this purpose. The *visualization interface* component provides web-based user interfaces to monitor IoT data at the edge continuously. The *notification/alert* component communicates with users through email or push notification services.

The *data transfer* component represents the ability of the edge device to send data across different networks to its cloud server or other systems. Network interfaces of the edge device and communication protocols need to be considered. The edge device, such as *Raspberry Pi*, has enabled diverse network interfaces. Wi-Fi, Ethernet, and 5G cellular are

standard network interfaces used to connect edge devices to cloud servers. Communication protocol services consist of the *publish–subscribe* and *request–response* messaging models. *MQTT* communication is the most popular *publish–subscribe* protocol for IoT application systems. It can operate on an edge device with limited processing power and memory. *HTTP* communication is often used for the *request–response* messaging model. In addition, the standardization format of data transfer should be addressed for this component. In this case, the JSON format is utilized.

The *action* component consists of functions that send commands to actuators through connectivity interfaces. Due to the complexity of action functions becoming more diverse, it should be able to execute different action functions in parallel or sequentially. The *data access API* is another output component that should be considered in the edge layer. It provides a function to allow external systems to access local data through *HTTP* communication, which is relevant to the current IoT trends of cross-vendor capabilities and interoperability. Thus, it enables the development of complex IoT systems that utilize multiple vendor services simultaneously.

### 3.3.4. Other Components

For developing the edge device, we should consider additional components that are not included in the *input*, *processing*, and *output* components. These components are management, scheduling, security, local data storage, remote debugging, and dynamic configuration. The *management* component controls and monitors the lifecycle of the edge device. The *scheduling* component controls the time cycle for executing data streams in the edge device. The *security* component provides privacy and security capabilities of the edge device.

When sensor data cannot be transmitted to the server, the system must provide the reliable local data storage service to archive sensor data records. The local data storage component should consider the battery consumption, latency, and CPU utilization. The lightweight embedded database engine, such as *SQLite*, can be the suitable database option with which to develop this component.

Currently, the edge management system provides dynamic configuration capabilities. It allows users to modify edge system parameter settings by changing environments. Parameter settings include connected sensors and actuators, data processing methods, and data transmission services. However, this component may cause problems and errors if the configuration does not match the current environment of the edge device. Therefore, the remote debugging component will be the solution. It allows running and verifying the new device configuration without affecting the existing system running on the edge device.

### 3.4. Cloud Layer

The cloud layer components are responsible for processing, analyzing, managing, storing, and visualizing IoT data using cloud-based services. These components perform computations that are not feasible on edge devices. In this paper, we present the cloud layer components in Figure 1. We organized them into *input*, *processing*, *output*, and *other* components.

The input components provide the services to receive sensor data from different devices using different communication protocols. It consists of the IoT gateway and the data aggregator. The components contain a variety of data processing functions for IoT data stream processing, filtering, rules engine, data synchronization, and analytics, with plug-in function capabilities, where each function should be implemented as a standalone one to prevent system failures. The output part concerns the ability of the cloud system to provide capabilities for users or other systems to access IoT data. The output components may include visualization functions, notification/alert functions, REST API services, business application integrations, and IoT collaboration capabilities. The other components provide additional components that will support the main services of the cloud server.

They include management, data storage, device management, user authentication, and security components.

Figure 1 shows the components of the cloud layer in the edge device framework. It consists of the IoT gateway, the data aggregation, and the device management component. The *IoT gateway* provides communication protocol services such as *HTTP* and *MQTT* to receive data from edge devices. The *MQTT* broker service was implemented to enable *MQTT* communication. The *REST API* service was developed for accepting sensor data through *HTTP POST* communications. Additionally, the IoT gateway component should consider potential utilizations of communication protocols provided by other cloud service providers.

The data input process at the cloud layer usually starts when the *IoT gateway* receives sensor data. It will be followed by *data aggregation*. Then, data will be forwarded to data processing functions and be stored in the data storage. The *data aggregation* component collects data from several data sources, applies data processing, and reassembles data in a usable format.

In this paper, we emphasized the importance of the *device management* component in the development of the edge device framework. The component manages the devices in the cloud system. It identifies device specifications, such as sensors that are connected to the edge device, and handles the integration between edge devices and the cloud server. It allows the dynamic configuration component of the edge to be triggered remotely from the cloud server. The device management data are stored in cloud server data storage.

## 4. *SEMAR* IoT Application Server Platform

In this section, we introduce *SEMAR* as an IoT application server platform to facilitate the development of a cloud layer system. In previous studies, we designed and implemented the *SEMAR* IoT application server platform in consideration of the cloud layer for the general IoT architecture described in Section 3. The current implementation of *SEMAR* has been used in several IoT application systems [30]. It provides the integration functions of collecting, displaying, processing, and analyzing sensor data, including *built-in* and *plug-in* functions. Figure 2 shows the system overview of the *SEMAR*.

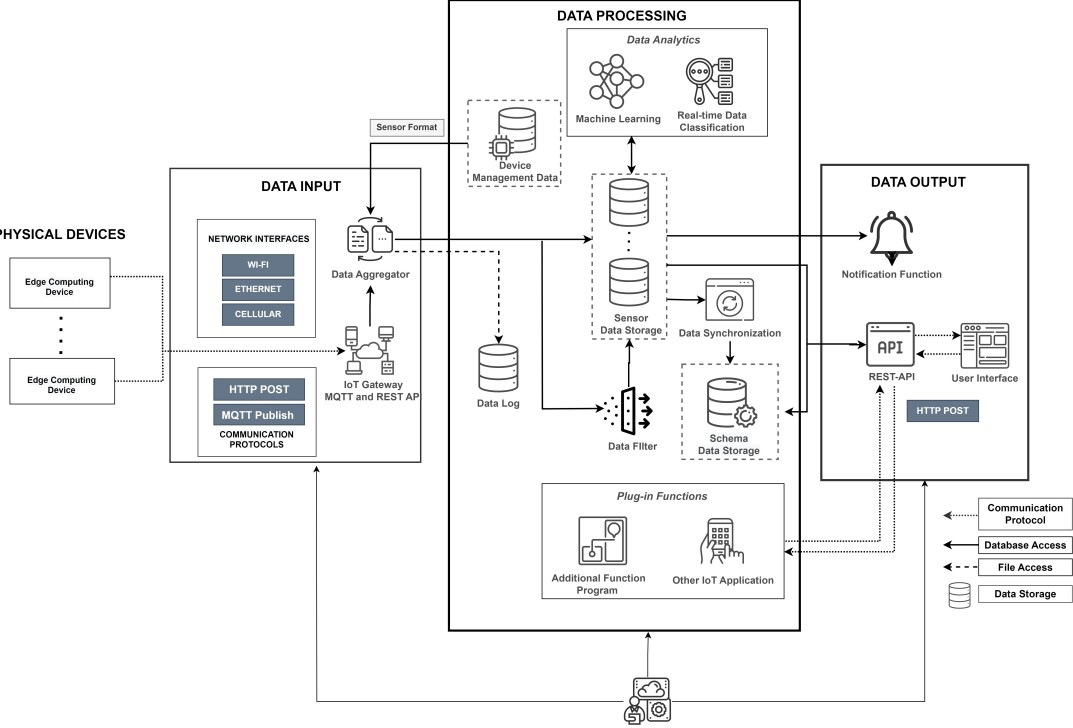

**Figure 2.** Design overview of *SEMAR* IoT application server platform.

The *built-in* function allows the use of new functions without implementing or modifying the original source codes. The components of the *built-in* function are grouped according to *data input*, *data processing*, and *data output* that are controlled by the *management* system.

The *data input* provides components for gathering sensor data from various IoT resources that accept connection through network interfaces and communication protocols. It consists of the IoT cloud gateway for communication services through *HTTP POST* and *MQTT* communication protocols, and the *data aggregator* for gathering and processing sensor data with the consumable format based on the sensor format stored in the device management data. It transmits the results to the *data processing* component and stores them in the *MongoDB* data storage [31].

The *data processing* components consist of the *data filter* for reducing noises and inaccuracies in the data obtained, the *data synchronization* for synchronizing the data from different devices and storing it in the dynamic database called the *schema data storage*, and the *data analytics* for analyzing large amounts of data.

The last component employs machine learning techniques and real-time data classification services. The machine learning techniques enable the user to construct a data model for the real-time data categorization feature using sample data from the data storage. In addition, the *SEMAR* IoT application server platform enables *plug-in* functions that can be implemented as system extensions or as the other IoT application systems to access the data through *REST API* services.

The *SEMAR* IoT application server platform includes several output components. Users can access the sensor and synchronized data through the user interface based on a website. It enables the data export function to download sensor data in CSV, JSON, Excel, or text format at a specific time by accessing the user interfaces. The notification function enables the user to set the threshold for each sensor data point as the message notification trigger. If the value fulfills the threshold, the system will generate and send an alert to users. In accordance with the current trend of IoT platforms, we implemented capabilities that enable IoT collaboration, which allows for the connections and integrations of other systems. We used the *REST API* service for data integrations and exchanges through *HTTP POST* communications using the JSON format. The *REST API* retrieved data from storage and translated it into the JSON format.

The *management service* has been implemented in the *SEMAR* to manage user authentications, devices data, and its communication protocol. In this paper, we improved the device management feature by adding the function to create, update, and delete the edge configuration file for the edge device. Users are able to operate and monitor the device remotely. Users can access this service through the user interface.

The procedure for integrating the *SEMAR* platform with a new IoT application system is described as follows:

- The user registers the devices and the sensors of the IoT application system on the *SEMAR* platform;
- The system prepares the IoT cloud gateway services, including the *HTTP POST* and *MQTT* communication protocols, to receive data;
- The device sends data to the server through the defined communication service in JSON format;
- The data are received by the IoT cloud gateway, processed by the *data aggregator* based on the registered sensors, and are stored in the *sensor data storage*.
- The *SEMAR* provides the capability of synchronizing data from several devices by accessing the *sensor data storage* and storing the results in the *schema data storage*;
- The user interface of *SEMAR* displays the data. The user can integrate their programs as *plug-in* functions by utilizing the *REST API*.

## 5. Design and Implementation of Edge Device Framework

In this section, we present the design and implementation of the edge device framework.

### 5.1. System Overview

The following section presents the edge device framework as a collection of tools that will make it easier to create edge computing systems. Figure 3 provides the overview of the integrated system of the edge device framework in *SEMAR*. It functions in three phases. In the *initialization phase*, it offers web services that enable the automatic downloading of the configuration file to the device via *HTTP* communications. In the *service phase*, it transforms data from various sensors into the standard data format and periodically transmits them to the server. In the *update phase*, it remotely updates the configuration through *MQTT* communications.

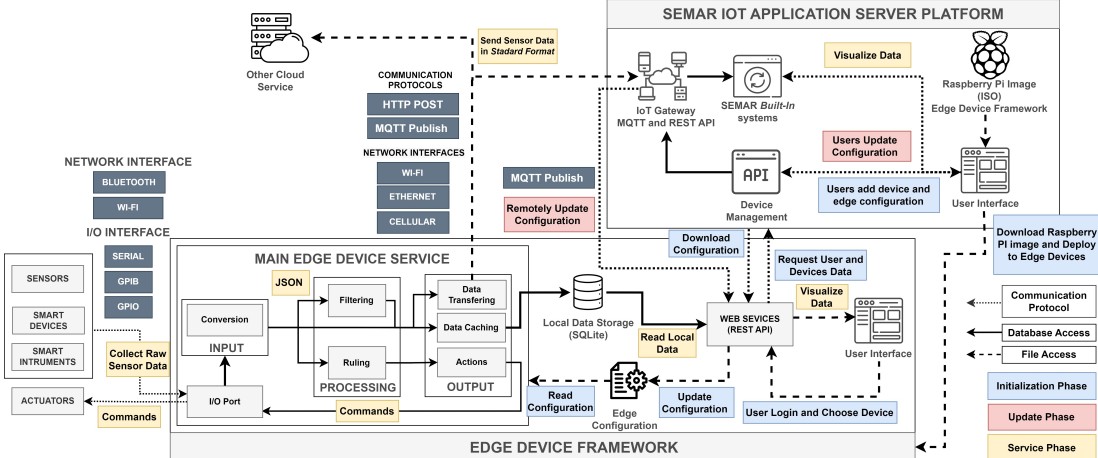

**Figure 3.** Design overview of the edge device Framework.

### 5.2. Initialization Phase

In the *initialization* phase, the framework is installed on the edge device, and the initial connection is established between the edge device and the *SEMAR* platform. First, the user registers a new device and configures the edge device on the *SEMAR* platform via the user interface. Then, the user downloads the *Raspberry Pi* image from the *SEMAR* platform and deploys it to the edge devices. The user needs to ensure that the devices are connected to the Internet. Next, the user accesses the web services of the edge device framework through the user interface. The system verifies the user account by accessing the *REST API* services of the *SEMAR* platform. If the user account is authenticated, the system retrieves all the device data of the user from the *SEMAR* platform, generates the *edge ID* of the device, and grants the access to the web services.

In the *initialization* phase, the user needs to choose the data to be applied to the edge device from the user interface. Then, the system downloads the edge configuration, saves it to the JSON file, and runs the *main service* program. Algorithm 1 illustrates the process flow of this program for both the *initialization* and *update* phases. Figure 4 shows the sample edge configuration file used in the framework. It includes the device, the device identity, and the configuration parameters such as the sensor interface, the data conversion method, the data model, transmitted data, the local data storage, and the local visualization. The required libraries to run the system have been installed in the edge device framework.

---

**Algorithm 1** Edge configuration service.

---

**Input** : Edge ID (*edgeID*)
**Output**: Edge configuration file (*EdgeConfig*)
**begin**
    Set *EdgeConfig* ← read *EdgeConfig* from the "*config.json*"
    **if** *EdgeConfig not NULL* **then**
        Run Main Service program(*EdgeConfig*)
        Connect to the *MQTT* broker in *SEMAR*
        Subscribe for the "*edgeID*" *MQTT* topic
        **while** *true* **do**
            **if** *Message ← receive data from server through MQTT communication* **then**
                Set *EdgeConfig* ← convert *Message* to JSON format
                Save *EdgeConfig* to the "*config.json*"
                Restart Main Service program(*EdgeConfig*)
            **end**
        **end**
    **end**
**end**

---

```json
1   {
2       "device_code": "ih37",
3       "configuration_code": "cn37",
4       "resource": "Data Logger [GL240]",
5       "interface": [{
6           "type": "wlan",
7           "config":{...},
8           "method": "web_scrapping",
9           ...
10          "object_used": {"CH 1": "CH 1", "CH 2": "CH 2"}
11      }],
12      "data_transmitted":{"ch_1": "CH 1","ch_2": "CH 2"},
13      "time_interval": 5,
14      "communication_protocol": {"mqtt": {"server": "103.106.72.181", "port": "1883", "topic": "sensor/logger"}},
15      "local_data": { "ch_1": ["CH 1", "real"],"ch_2": ["CH 2", "real"]},
16      "visualization": {
17          "table": ["ch_1", "ch_2"],
18          "graph": [{"value": [{"title": "Channel 1","field": "ch_1"},{"title": "Channel 2","field": "ch_2"}]}]
19      }
20  }
```

**Figure 4.** A sample of the edge configuration file in JSON format.

### 5.3. Service Phase

In the *service phase*, which is the primary phase of the edge device framework, the framework collects and transmits sensor data to *SEMAR*. Figure 3 illustrates the lifecycle of the edge device framework for this purpose. Based on the general IoT application architecture illustrated in Figure 1, the functions of the main edge framework services are classified into *data input*, *data processing*, and *data output*. Algorithm 2 describes the program flow. To collect the raw sensor data, the edge device must be connected to the sensor or device. The service program then reads the edge configuration file, which was downloaded by the edge configuration services. The program can process the raw sensor data by converting them to the standard data format, reducing inaccuracies in the data using the *filtering* function, generating the decisions based on predefined rule models using the *ruling* function, saving it to local data storage, and sending it to the server in JSON format using a defined communication protocol. The communication protocol can be either *MQTT* or *HTTP POST*. The *SEMAR* platform receives, processes, and analyzes the sensor data using built-in systems on the server and displays the sensor data as output in the user interface. Additionally, the system can send notifications/alerts to the user and trigger actuators based on the rule model results. The program runs periodically at specific intervals and only transmits the sensor item values defined in the configuration file. Therefore, the framework enables the user to manage edge devices and optimize their performance by defining edge configuration files.

---

**Algorithm 2** Service phase.

---

**Input** : Edge configuration(*EdgeConfig*)

**begin**

    Set *TimeInterval*, *CommService*, *Interface*, *TransmitData*, *FilterModel*, *RuleModels*, *LocalData*, *ActionModels* ← read the configuration of time interval, communication service, resource interface, transmitted data from *EdgeConfig*

    Set *SensorResource* ← connect to the network interface of sensor device(*Interface*)

    **while** *true* **do**

        Set *RawSensor* ← read raw data of sensor from *SensorResource*

        Set *ConvertData* ← convert raw data of sensor to the standard format(*RawSensor*, *Interface*)

        **if** *FilterModel not empty* **then**

            Set *ConvertData* ← procces sensor data using digital filter(*ConvertData*,*FilterModel*)

        **end**

        **if** *RuleModels not empty* **then**

            Set *RullingResults* ← applying rule models(*ConvertData*,*RuleModels*)

        **end**

        Save sensor data to the local storage(*ConvertData*,*LocalData*)

        Set *Data* ← select transmitted sensor data(*ConvertData*, *TransmitData*)

        Send transmitted data to the server through communication service (*Data*, *CommService*)

        **if** *RullingResults not empty* **then**

            Send commands to control actuators(*RullingResults*,*ActionModels*)

        **end**

        sleep(*TimeInterval*)

    **end**

**end**

---

One difficulty in inputting data into the edge device framework involves the connectivity of the sensor interface. The aim of the edge device framework is to create a versatile edge computing device that can automatically gather and transmit sensor data to the server. Therefore, it is essential to establish connectivity services and data models that can support multiple sensors. Currently, our system can capture and transform sensor data through the GPIO, USB serial, and wireless interfaces. We have created multiple functions with which to collect data from the GPIO interfaces. To use the system, the user must first specify the GPIO ports and modes in the configuration file. Then, the system periodically reads the port value, converts it into a JSON object based on the configuration file, and returns the results to the data processing components.

To use the USB serial interface, the user needs to specify the serial port, the timeout time, and the baud rate that determines the data transmission speed. The user also needs to define the delimiter that the system will use to extract the relevant information when it receives a line of serial communication data. Algorithm 3 shows the data conversion process for serial communications.

---

**Algorithm 3** Data conversion procedure for serial communication.

---

**Input** : Raw sensor data (*RawSensor*), Edge configuration(*EdgeConfig*)
**Output:** Converted sensor data (*ConvertData*)
**begin**
    Set *Delimeter*, *ObjectUsed* ← read configuration of delimiter and object used, from *EdgeConfig*
    Initialize *ConvertedData*, *Result* ← empty JSON object
    Set *DataList* ← SPLIT(*RawSensor*, *Delimeter*[0])
    **for** *each item in DataList* **do**
        Set *Buffer* ← SPLIT(*item*, *Delimeter*[1])
        Set *Result*[*Buffer*[0]] ← *Buffer*[1]
    **end**
    **for** *each sensor in ObjectUsed* **do**
        **if** *sensor in Result* **then**
            Set *ConvertData*[*sensor*] ← *Result*[*sensor*]
        **end**
    **end**
    **return** *ConvertData*
**end**

---

To use the wireless interface, the user needs to provide the URL of the web service to receive the HTML data through *HTTP GET* communications. The web scraping technique is used to extract the necessary information from the HTML data and to transform it into an array format. The user needs to define the index array that includes the channel name and sensor value. The data conversion process for the wireless interface data is shown in Algorithm 4, which illustrates the data conversion procedure for the wireless interface data.

---

**Algorithm 4** Data conversion procedure for wireless interface.

---

**Input** : Raw sensor data in HTML format (*RawSensor*), Edge configuration(*EdgeConfig*)
**Output:** Converted sensor data (*ConvertData*)
**begin**
    Set *ChannelIndex*, *ValueIndex*, *MaxSequence*, *ObjectUsed* ← read configuration from *EdgeConfig*
    Initialize *ConvertedData*, *Result* ← empty JSON object
    Set *DataList* ← WEBSCRAPING(*RawSensor*)
    **for** $i \leftarrow 0$ *to length*(*DataList*) **do**
        **if** $i \% MaxSequence == ChannelIndex$ **then**
            Set *ChannelName* ← *DataList*[*i*]
        **end**
        **if** $i \% MaxSequence == ValueIndex$ **then**
            Set *SensorValue* ← *DataList*[*i*]
        **end**
        **if** $i \% MaxSequence == ( Maxsequnce - 1 )$ **then**
            Set *Result*[*ChannelName*] ← *SensorValue*
        **end**
    **end**
    **for** *each sensor in ObjectUsed* **do**
        **if** *sensor in Result* **then**
            Set *ConvertData*[*sensor*] ← *Result*[*sensor*]
        **end**
    **end**
    **return** *ConvertData*
**end**

---

The data transfer system has been implemented to enable the transmissions of sensor data to not only the *SEMAR* platform but also to any other IoT gateway service the user prefers for the cross-vendor capability in edge computing. It currently supports *HTTP POST* and *MQTT* communications using the standard JSON format. The data transfer

function uses the *"time_interval"* configuration to regulate the data transfer frequency, the *"data_transmitted"* configuration to determine the output data to be transferred, and the *"communication_protocol"* configuration to describe the destination and communication service. While developing edge devices, the communication network is the critical factor for avoiding the unsuccessful data transfer. The data caching function is implemented by using SQLite and Python to store sensor data locally, with the *"local_data"* configuration specifying which data are saved in the local data storage.

The current implementation allows the user to visualize data in the forms of tables and graphs. It is accomplished using the *"visualization"* setting, which retrieves sensor data from an *SQLite* database. The user can access these data through the web interface or the *REST API* service. To make IoT application system developments more flexible, we suggest the use of the *REST API* service at the edge layer to integrate edge device frameworks with other systems.

### 5.4. Update Phase

In the *update phase*, the user has the ability to remotely modify the edge configuration file on the edge device using the *SEMAR* user interface. This process involves modifying the edge configuration and utilizing the deploy button to initiate the remote update function. The *device management* service transmits the updated edge configuration in the JSON format to the relevant edge device using *MQTT* communications with the edge ID as the topic. The edge configuration service connects to the *MQTT* broker within *SEMAR* and subscribes to the same topic with the edge ID. After receiving the new edge configuration through *MQTT* communications, the service saves it in the designated folder and triggers the function to restart the service program. As a result, the user can easily add new sensor devices or modify device configurations by making adjustments through the user interface. Figure 5 illustrates the flow process of the *update phase*.

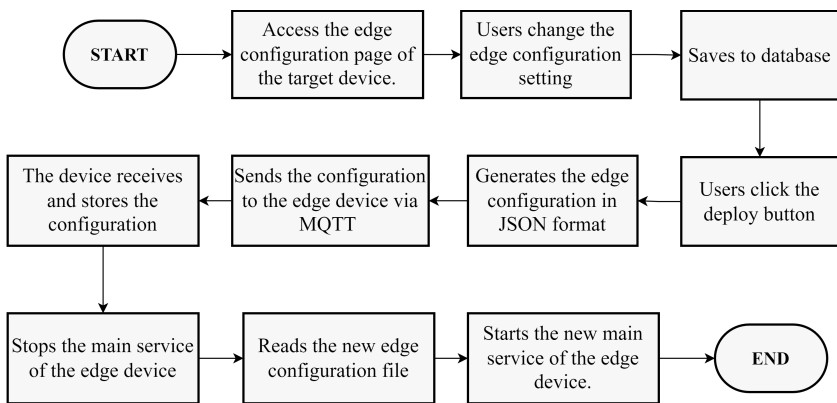

**Figure 5.** Flow diagram of *update phase*.

## 6. Application for Fingerprint-Based Indoor Localization System

As the first application, we integrated the *FILS15.4* into the *SEMAR* IoT application platform [30]. This system is used to detect the user locations in indoor environments based on the fingerprints of the target location. The procedure consists of a *calibration phase* and a *detection phase* [16,17].

### 6.1. System Architecture

Figure 6 illustrates the overview of the *FILS15.4* architecture. The *FILS15.4* system utilizes the transmitting and receiving devices produced by *Mono Wireless* that operate on the *IEEE802.15.4* standard at 2.4 GHz [32]. The transmitter *Twelite 2525* has the dimensions of 2.5 × 2.5 cm, and is powered for a long time by a coin battery. The receiver *Mono Stick* is connected to the *Raspberry Pi* through a USB connection.

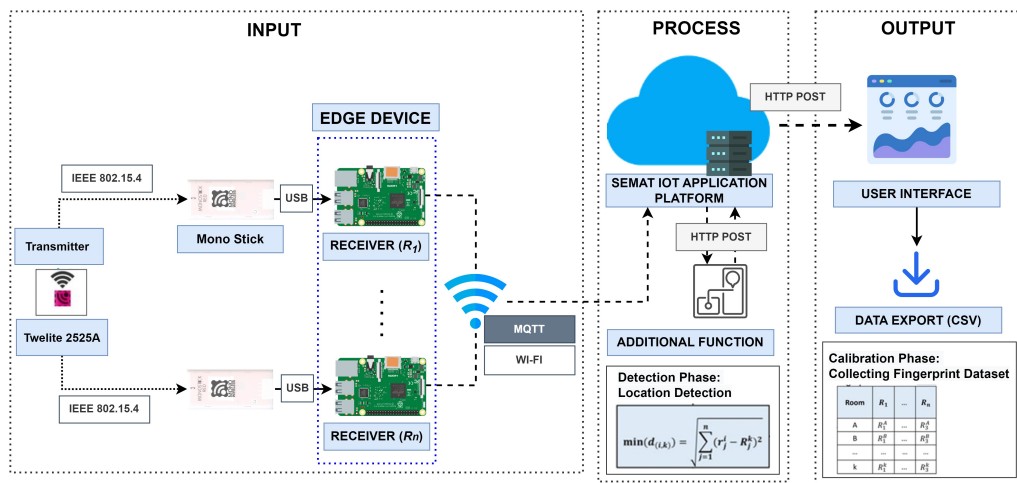

**Figure 6.** System overview of *FILS15.4*.

*Raspberry Pi* collects the data from a transmitter by receiving data at the *Mono Stick* through USB serial communications. It determines the *link quality indication (LQI)* for each transmitter and sends the data consisting of the LQI value and the transmitter ID to the server through the *MQTT* communication protocol. The server receives the data, synchronizes the data from all the receivers by calculating the average LQI with the same transmitter ID, and stores the results in one record in the database.

### 6.2. Calibration Phase

The *calibration phase* produces and records the fingerprint dataset. Each fingerprint consists of *n* LQI values, where *n* represents the number of receivers. It indicates the features of LQI values when a transmitter is placed at the specified location (room in *FILS15.4*).

### 6.3. Detection Phase

The *detection phase* identifies the current location of the transmitters by measuring the *Euclidean distance* between the current LQI values from the receivers and the fingerprint dataset for each room stored in the database and selecting the fingerprint with the shortest distance.

### 6.4. Evaluation of Implementation

The implemented edge device framework for *FILS15.4* was deployed on two floors in the #2 Engineering Building at Okayama University for evaluations. Our evaluations intended to verify the adaptability and the validity of the edge device framework in *SEMAR*. Table 1 presents the device and software specifications for this evaluation.

**Table 1.** Device and software specifications of *FILS15.4*.

| Components | Items | Specifications |
|---|---|---|
| Edge Device | Model | Raspberry Pi 4B |
| | Operating System | Linux Raspbian |
| Sensor Device | Model | Twelite Mono Stick |
| | Sensor Interface | USB |
| | Communication Method | Serial Communication |
| | Collected Data | *id, lqi, accelero x, y and z* |

We evaluated the ability of the edge device framework to automatically install the edge configuration built on *SEMAR* to the edge device, collect sensor data, convert them, and send them to the server by following the configuration file. In addition, we evaluated

the configuration update feature by modifying the edge configuration setting and remotely deploying it to the edge device through the user interface of *SEMAR*.

Figure 7 shows the initial configuration file for *FILS15.4*. The interface includes the configuration of the serial communication for collecting data from a USB receiver and the parameter for obtaining the necessary data by converting them to the standard format. According to the edge configuration, the system sends sensor data that consist of *ID*, *LQI*, and *accelerometer x, y, z*, to the server at every 0.5 s (500 ms) through the *MQTT* communication.

```
1    {    "resource": "USB Mono Stick",
2        "interface": [{
3            "type": "usb_serial",
4            "config": {"port": "/dev/ttyUSB0","baudrate": 115200,"timeout": 1},
5            "string_pattern": "rc=[rc-value]:lq=[lq-value]:ct=[ct-value]:ed=[ed-value]:id=[id-value]:ba=[ba-value]
             :a1=[a1-value]:a2=[a2-value]:x=[x-value]:y=[y-value]:z=[z-value]",
6            "delimeter": [":", "="],
7            "object_used": {"id": "id","lq": "lq","x": "x","y": "y","z": "z"}
8        }],
9        "data_transmitted": {"id": "id","lqi": "lq","x": "x","y": "y","z": "z"},
10       "time_interval": 0.5,
11       "communication_protocol": {"mqtt": {"server": "103.106.72.181","port": "1883","topic": "sensor/edge/fils"}}
12   }
```

**Figure 7.** Edge configuration for receiver device of *FILS15.4*.

Figure 8 illustrates the updated edge configuration for *FILS15.4*. It is changed from the initial configuration. The configuration was modified by removing the accelerometer data from the result of the data converter process, and only transmitting *ID* and *LQI* data to the server. The data transmission interval is similar to the previous configuration.

```
1    {    "resource": "USB Mono Stick",
2        "interface": [{
3            ....
4            "object_used": {"id": "id","lq": "lq"}
5        }],
6        "data_transmitted": {"id": "id","lqi": "lq"},
7        ...
8    }
```

**Figure 8.** Updated edge configuration for receiver device of *FILS15.4*.

Figure 9 shows the data visualization of *FILS15.4* through the *SEMAR* user interface. The initial configuration part indicates that the edge device can collect data from the USB receiver, convert them, and send them to the server by following the initial configuration in Figure 7. The updated configuration part represents the edge device when it collects, processes, and transmits data by following the updated configuration in Figure 8.

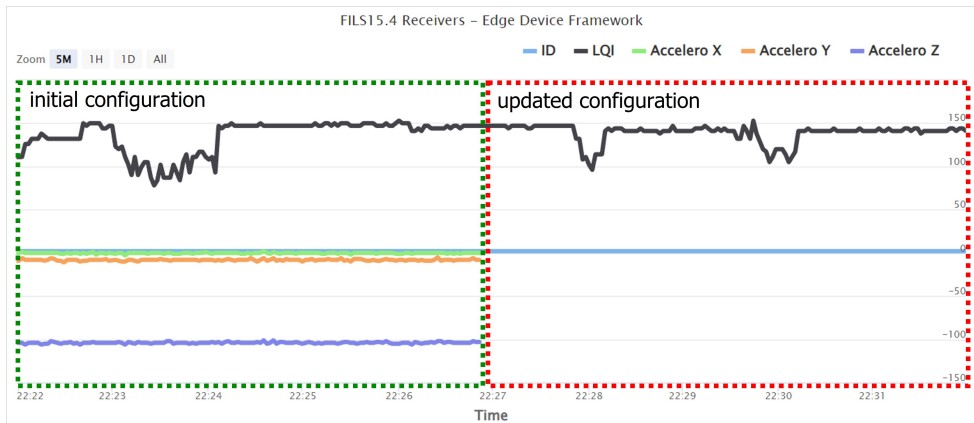

**Figure 9.** Data visualization of the *FILS15.4* receiver device.

## 7. Application for Data Logging System

As the second application, the data logging system is integrated to enable real-time monitoring of the temperature data of some materials during the quenching heat treatment process.

### 7.1. System Overview

Figure 10 illustrates the overview of the *data logging system* architecture. This system uses *midi Logger GL240* with *WLAN B-568* that is provided by *Graphtec* [33] to capture the temperature data during the quenching heat treatment process by attaching the sensor to the material. The treatment process is used for hardening steel by putting the material into the heater machine to improve metal performances. *WLAN B-568* provides the *HTML* web service for displaying the data collected by the data logger. The integration of the data logger with the IoT application server platform is as follows:

- The edge device for the data logging system captures raw sensor data in the HTML format by accessing the data logger web services through wireless communications;
- It reads the input HTML data, extracts the temperature value using web scraping techniques, and transforms it into JSON format;
- It transmits the JSON data to the *SEMAR* platform through the *MQTT* communication protocol;
- The *SEMAR* platform receives, processes, and saves the sensor data in the database;
- The *SEMAR* platform displays the sensor data through the user interfaces.

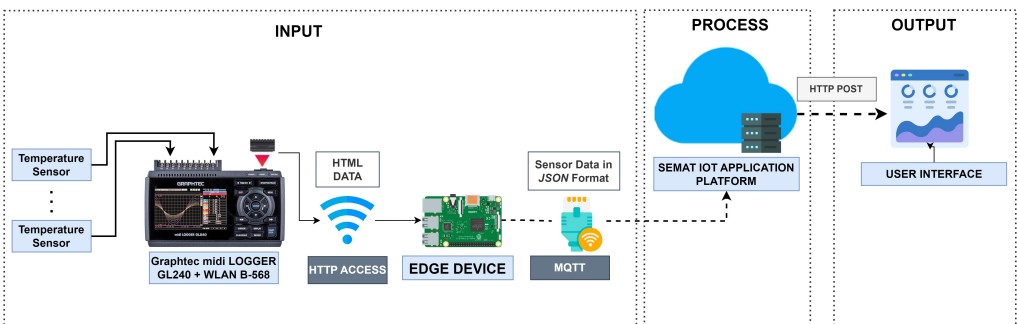

**Figure 10.** System overview of *data logging system*.

### 7.2. Evaluation of Implementation

We evaluated the implementation of the edge device framework for the *data logging system* by running it in the #1 Engineering Building at Okayama University. Our evaluations intended to verify the adaptability and the validity of the edge device framework in *SEMAR*. Table 2 presents the device and software specifications for this evaluation.

**Table 2.** Device and software specifications for *data logging system*.

| Components | Items | Specifications |
|---|---|---|
| Edge Device | Model | Raspberry Pi 4B |
| | Operating System | Linux Raspbian |
| Sensor Device | Model | midi Logger GL240 and WLAN B-568 |
| | Sensor Interface | Wireless Connection |
| | Wireless LAN Mode | Access Point |
| | Wireless LAN IP | *192.168.230.1* |
| | Web Services URL | http://192.168.230.1/digital.cgi?chgrp=0 (accessed on 19 December 2022) |
| | Communication Method | *HTTP* Communication |
| | Collected Data | *temperature* |

Figure 11 shows the initial configuration file for the *data logging system*. The edge configuration indicates that the edge device collects data from the data logger through the wireless network. It transmits the measured temperature data from *channels* 1 and 5 to the server every five seconds through the *MQTT* communication.

```
1    {    "resource": "Data Logger",
2        "interface": [{
3            "type": "wlan",
4            "config": {"url": "http://192.168.230.1/digital.cgi?chgrp=0","timeout": 8},
5            "method": "web_scrapping",
6            "index_name": 0,
7            "index_value": 1,
8            "max_sequence": 3,
9            "object_used": {"ch_1": "CH 1","ch_2": "CH 2","ch_3": "CH 3","ch_4": "CH 4","ch_5": "CH 5","ch_6": "CH
                 6","ch_7": "CH 7","ch_8": "CH 8","ch_9": "CH 9","ch_10": "CH 10"
10           }
11       }],
12       "data_transmitted": {"ch_1": "ch_1","ch_5": "ch_5"},
13       "time_interval": 5,
14       "communication_protocol": {"mqtt": {"server": "103.106.72.181","port": "1883","topic": "sensor/logger2"}}
15   }
```

**Figure 11.** Edge configuration for edge device in the data logging system.

Figure 12 shows the updated edge configuration of the data logger monitoring system. It was modified from the initial configuration. In this configuration, the transmitted data were changed by only sending the temperature data from *channel* 1 every 2 s through the *MQTT* communication.

```
1    {    "resource": "Data Logger",
2        "interface": ....,
3        "data_transmitted": {"ch_1": "ch_1"},
4        "time_interval": 2,
5        "communication_protocol": ....
6    }
```

**Figure 12.** Updated edge configuration for edge device of data logging system.

Figure 13 illustrates the data visualization of the data logging system. The initial configuration part represents the edge device for collecting, processing, and transmitting data according to the initial configuration in Figure 11. Additionally, the updated configuration part shows the data sent by the edge device when the configuration is modified according to Figure 12.

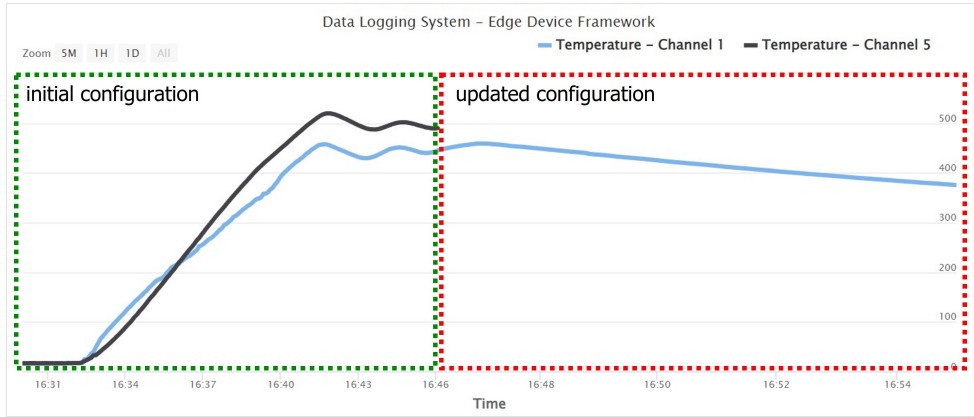

**Figure 13.** Data visualization of the data logging system.

## 8. Evaluations

In this section, we evaluated the implementation of the *SEMAR* IoT server platform.

### 8.1. Performance Analysis

The first evaluation of the edge device framework's performance involved investigating the average CPU and memory usage of the main service program while collecting and transmitting sensor data at various time intervals. This evaluation was crucial for assessing the computational performance of the framework during the main phase. To carry out this evaluation, we employed the data logging system application and measured the average memory and CPU usage during the experiment time as shown in Figures 14 and 15. We tested different time intervals ranging from 0.1 s to 10 s for three minutes each and utilized the feature described in Section 5.4 to modify the time interval configuration.

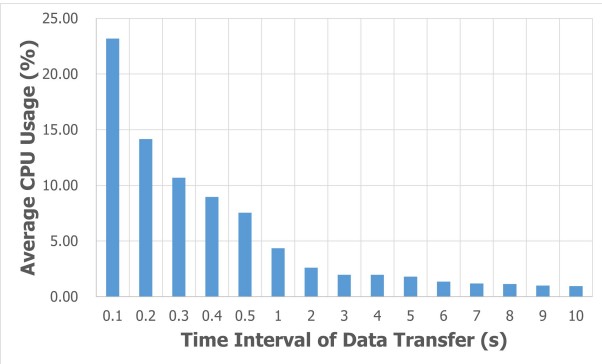

**Figure 14.** Average CPU usage rate of main services with different time intervals.

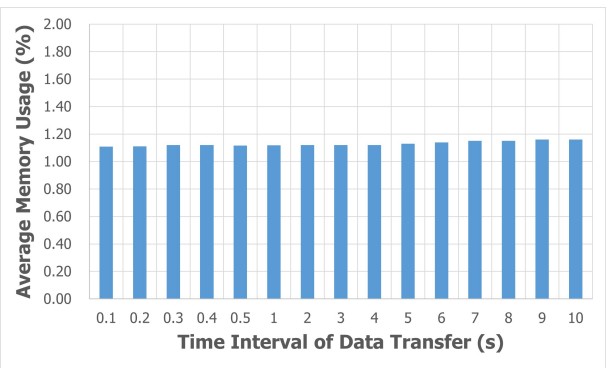

**Figure 15.** Average memory usage of main services with different time intervals.

The second evaluation involved examining the average response time of the web services when accessed by multiple users simultaneously via *HTTP POST* communications. The edge device framework was installed on a *Raspberry Pi*, and a considerable amount of sensor data were stored. To simulate multiple users, we developed a simulation program that generates virtual users, and ran it on personal computers connected to the *Raspberry Pi* via Ethernet in the local area network. During the experiments, we increased the number of user accesses from 5 to 150, with each virtual user representing an actual user or system using the device data. All the virtual users used similar parameter requests to access sensor data stored in local data storage.

To measure the response time, we calculated the time difference between the case where a virtual user sends a request to the web services and the case where it receives the response message. The response message is the 56 KB JSON message containing 500 records of data. During the experiment, we also evaluated the throughput of web services, which was 2.3 MB/s. It can handle 41 requests per second. Figures 16 and 17 illustrate the average response time and the average CPU usage rate, respectively, when the number of virtual users increased from 5 to 150.

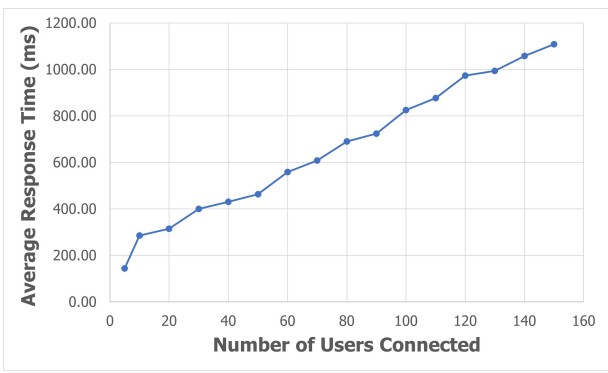

**Figure 16.** Average response time of web services with different numbers of users connected.

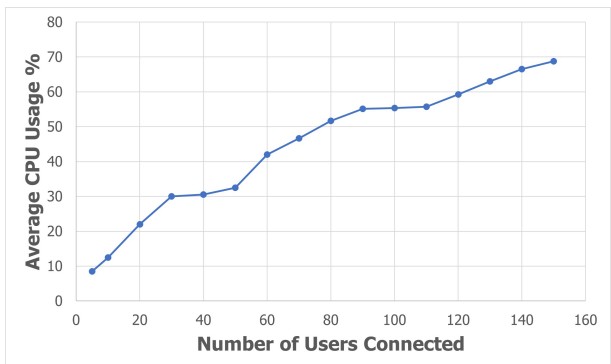

**Figure 17.** Average CPU usage rate of web services with different numbers of users connected.

### 8.2. Comparative Analysis

We compared features of the edge device framework with eight research works taking similar approaches in the literature. We compiled a list of features to be considered for comparing different edge computing systems frameworks. They were used to characterize each proposal, and included the following:

- The *main purpose* was to identify the issue that the proposed system intends to address and the key reason for selecting it to run edge IoT applications.
- *Edge devices* represent devices that installed an edge computing framework system.
- *Dynamic deployment* shows the ability to allow users to dynamically configure the flow system to run their own edge applications based on hardware and process requirements (Yes or No).
- *Remotely update* indicates the capability to remotely update the system (Yes or No).
- *Data conversion* implies the capability to preprocess data across several devices into a standard format (Yes or No).
- *Scalability* demonstrates the ability to expand their applications and to execute the number of data processing requests simultaneously (Yes or No).
- *Interoperability* indicates the capability to connect through several widely adopted and supported protocols provided by multiple devices (Yes or No).
- *Cross-vendor capabilities* illustrate the capacity of edge computing to collaborate with multiple vendors to develop complex IoT application platforms (Yes or No).

Table 3 compares the fulfillment of the eight features among the eight related works and our proposed edge devices framework.

**Table 3.** The comparative evaluation between the proposed framework and the existing related studies.

| Work Reference | Main Purpose | Edge Devices | Dynamic Deployment | Remotely Update | Data Conversion | Scalability | Interoperability | Cross-vendor Capabilities |
|---|---|---|---|---|---|---|---|---|
| [34] | Data stream processing and task management | Wi-Fi Home Gateway | ✓ | ✗ | ✗ | ✓ | ✓ | ✗ |
| [35] | Edge devices gateways and support tool | Personal Computer and Server | ✓ | ✓ | ✗ | ✓ | ✓ | ✓ |
| [36] | Edge devices for smart manufacturing | Single-Board Computer | ✓ | ✓ | ✓ | ✓ | ✓ | ✗ |
| [37] | Edge framework for smart farming | Personal Computer | ✓ | ✗ | ✗ | ✓ | ✓ | ✗ |
| [38] | Edge computing gateways | Server | ✓ | ✗ | ✗ | ✓ | ✓ | ✓ |
| [39] | Edge computing framework | Personal Computer | ✓ | ✓ | ✓ | ✓ | ✓ | ✗ |
| [40] | Edge devices for smart home | Personal Computer | ✓ | ✗ | ✗ | ✓ | ✓ | ✗ |
| [13] | Edge computing framework | Single-Board Computer | ✓ | ✓ | ✗ | ✓ | ✓ | ✗ |
| Our Proposal | General edge computing framework | Single-Board Computer | ✓ | ✓ | ✓ | ✓ | ✓ | ✓ |

### 8.2.1. Overview

Sajjad et al. in [34], Banerjee et al. in [35], and Ullah et al. in [38] developed systems that are consistent with the main objective of the edge computing framework by collecting data from diverse devices. Moreover, Rong et al. in [39] and Berta et al. in [13] created an edge computing framework that can gather data and connect to the actuator as the system output, which is similar to our edge device framework. Our framework is a general framework for edge computing and has the ability to connect with several IoT networks and to offer multiple output components that utilize the acquired data.

### 8.2.2. Edge Devices

Multiple works have used personal computers for installing and operating the frameworks. Nevertheless, they do not support the GPIO connectivity that is commonly used in sensor devices. Chen et al. in [36] and Berta et al. in [13] have implemented framework systems using single-board computer devices, such as the *Raspberry Pi*, which has significant benefits. Hence, we chose to deploy the proposed framework on these devices. This approach enables sensors to connect directly to the single-board computer devices for data collections, making the development of IoT application systems more straightforward.

### 8.2.3. Framework Features

In terms of framework features, all the related works offer capabilities for gathering data from IoT devices and sending them to a cloud server. However, as in our proposal, the works by Chen et al. in [36] and Rong et al. in [39] included the feature to process sensor data by converting them based on user-defined configurations.

All the works examined provided the capability to dynamically set up and deploy the framework using the connected devices as the main requirement. Some works required direct access to the devices for operations. Notably, Banerjee et al. [35], Chen et al. [36],

Rong et al. [39], Berta et al. [13], and our proposed framework allow users to remotely update the configuration from the cloud server.

### 8.2.4. Scalability, Interoperability, and Cross-Vendor Capabilities

All the works that have been reviewed focus on incorporating the scalability and interoperability in the functionality. However, some of them have the limited methods of connectivity for linking IoT devices to the edge framework. For instance, Sajjad et al.'s work [34] only allows the connectivity via Wi-Fi communications, whereas Zamora et al. [37] and Sharif et al.'s works [40] only permit connections from control unit devices to receive sensor data. Some works consider the cross-vendor capabilities of edge computing frameworks, particularly regarding data output components. Banerjee et al. [35], Ullah et al. [38], and the proposed framework allow the user to access to sensor data from edge devices using the *REST API*. However, only the proposed framework provides the additional features that allow data transmissions to various cloud computing vendors through *MQTT* and *HTTP POST* communications.

### *8.3. Discussions*

This sub-section outlines the performance evaluation outcomes of the proposed edge device framework in this paper. The framework was developed for the universal edge computing device with the primary objective of enhancing the effectiveness of building IoT application systems. The framework provides the flexibility to specify system configurations, such as time intervals for collecting and transmitting data to the server periodically. As the optimal time interval may vary depending on the purpose, we assessed the computing performance of the edge device framework across various time interval settings.

Figures 14 and 15 exhibit the mean CPU and memory usage during the execution of the main services across different time intervals. The results indicate that shorter time intervals require higher percentages of the CPU usage, where all the experimental results fall below 25%. Moreover, the amount of the memory usage remains relatively stable across time intervals, suggesting that the proposed system operates without demanding excessive computational resources.

The advent of *SIoT* has increased the complexity and universality of IoT application systems by enabling other services to access to sensor data beyond merely transmitting them to the server. As a result, it is crucial to include features that simplify data accessibility. To this end, we integrated web services that enable users to access sensor data via *HTTP* communications. Furthermore, we evaluated the communication and computing performance of the edge device framework when multiple users access it simultaneously.

Figures 16 and 17 illustrate the average response time and the CPU usage rate when the number of virtual users increases from 5 to 150. The average response time is 824 ms, and the CPU usage rate is 55% for 100 devices with the response message containing 500 data records. These results indicate that the proposed edge device framework can accommodate hundreds of users with the reasonable response time and the CPU usage rate.

To illustrate the latest developments in edge computing frameworks, we evaluated several comparable models and extracted relevant information from their published papers. Our analysis results, presented in Table 3, lead us to believe that the edge device framework offers advanced features and functionality that are highly valuable, especially given the growing trend towards developing more complex and general IoT application systems.

Furthermore, the evaluation results for the fingerprint-based indoor localization system and the data logging system demonstrate that the edge device framework can automatically retrieve the edge configuration file from the server. It can execute the service program by following the configuration file, gather data from the sensor, convert it to the standard format, and send it to the server within the pre-defined time frame using the communication protocol service. In addition, Figures 9 and 13 indicate that it allows users to remotely update the device configurations through the web interface of *SEMAR*. Hence, the implemented edge device framework in this paper can enhance the usage of

device sensors and contribute to the efficient development of IoT application systems utilizing *SEMAR*.

*8.4. Generalization*

According to the results of this paper, the *edge device framework* can improve utilizations of IoT devices by enabling users to remotely configure the parameters, including the connectivity of sensor interfaces, data conversions, data models, local data storage, local visualizations, and data transmission intervals to the server. All of the configuration parameters will be stored in the database to be used as templates for future use of similar sensors.

This framework was built on Python and can be utilized on any single-board computer supporting Python. It includes *NVIDIA Jetson Nano* [41], *BeagleBone Black* [42], *UDOO X86* [43], and *Odroid XU4* [44].

The *SIoT* architecture [25] was considered to develop a general edge computing device. Each IoT object in the *SIoT* architecture provides the computation and communication capabilities. The framework functions can be classified as *data input*, *data processing*, and *data output*. In *data input*, the functions to handle various sensor connectivities and data converters are implemented, including GPIO, GPIB, serial, and wireless communications. It offers multiple *data output* components to utilize the sensor data obtained. *REST API* data access is included for data transmissions to cloud services.

In the *MIoT* architecture [26], the function to manage the data communication model between IoT objects in static scenarios was implemented. This framework architecture includes the function that allows data communications in dynamic scenarios. It is possible to add new interactions between sensors and edge devices using *data input* components, and to manage data communications between the edge layer and the cloud layer through *data output* components, allowing cross-vendor data communications in the standard JSON format.

## 9. Conclusions

This paper presented the design and implementation of the *edge device framework* in the *SEMAR* IoT application server platform. It can remotely optimize device utilizations by configuring it through the *SEMAR* interface. The framework defines the connectivity of sensor interfaces, the data processing, the transmitted sensor elements, the communication protocol, the local data storage, the local visualization, and the data transmission interval on the server. It enables connection to a variety of sensor interfaces, transforms the data into a standard format, and provides multiple output components for data utilization.

Our evaluation results through applications with two IoT application systems verified the adaptability and validity of the proposed framework. IoT edge systems were developed in dynamic scenarios by allowing users to add or remove sensor devices flexibly.

In future works, we will continue enhancing the proposed framework, including implementations of the edge configuration validation function and the remote debugging function in *SEMAR*. They are necessary to prevent errors and guarantee consistency and reliability, and to find and fix problems in the edge systems. Then, we will continue to evaluate it through applications to other IoT application systems.

**Author Contributions:** Conceptualization, Y.Y.F.P., N.F. and S.S.; Methodology, Y.Y.F.P. and R.H.; Software, Y.Y.F.P., S.I. and J.S.; Writing—Original Draft Preparation, Y.Y.F.P.; Writing—Review and Editing, N.F.; Validation, M.K. and M.O. All authors have read and agreed to the published version of the manuscript.

**Funding:** This research received no external funding.

**Institutional Review Board Statement:** Not applicable.

**Informed Consent Statement:** Not applicable.

**Data Availability Statement:** Not applicable.

**Acknowledgments:** The authors thank the reviewers for their thorough reading and helpful comments.

**Conflicts of Interest:** The authors declare no conflict of interest.

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
