# Peer review of "An Edge Device Framework in SEMAR IoT Application Server Platform"

_information, doi:10.3390/info14060312_

Round 1

Reviewer 1 Report (Previous Reviewer 2)

In this paper, the authors propose an edge device framework for SEMAR, a platform for collecting, displaying and analyzing sensor data. The goal of the framework is to optimize edge device utilization.

The topic considered by the authors is interesting; in fact, both sensors and edge computing are hot topics in the current context of computer science.

The section on related literature is rich and comprehensive by also considering very recent architectures such as SIoTs and MIoTs.

The description of the proposed architecture is comprehensive and well organized. The description is done at the right technical level since the authors present overviews and then go into detail describing some algorithms.

The section on Application for Fingerprint-based Indoor Localization System is interesting and gives the reader an idea of a real case where the proposed framework can be applied. A similar argument applies to the Application for Data Logging System.

Finally, the section on experiments is interesting.

The quality of English is good.

Author Response

Dear Reviewer,
We again appreciate your kindness in reviewing our paper and providing valuable comments. Your insightful and valuable comments help improve the quality of the current version manuscript. All authors have reviewed and agreed to the submission of the revised manuscript. We hope that the revised manuscript meets your high standards.

We provide the point-by-point responses in our response letter.
Please see the attached file.

Reviewer 2 Report (Previous Reviewer 1)

The manuscript contains some details that are not relevant to the description of the framework and applications. For example, JSON device configuration is a well-known process for IoT developers. 

Little improvements are still needed throughout the paper.

Author Response

Dear Reviewer,
We again appreciate your kindness in reviewing our paper and providing valuable comments. Your insightful and valuable comments help improve the quality of the current version manuscript. All authors have reviewed and agreed to the submission of the revised manuscript. We hope that the revised manuscript meets your high standards.

We provide the point-by-point responses in our response letter.
Please see the attached file.

This manuscript is a resubmission of an earlier submission. The following is a list of the peer review reports and author responses from that submission.

Round 1

Reviewer 1 Report

The paper is well written and organized. The topic of IoT and edge computing/device is interesting but largely explored in the literature.

I believe, the presented contribution lacks originality and research content. The work is merely an IoT device onboarding scheme with an edge computing device downloading configuration from an IoT server and pushing sensor data to it. The scheme is well engineered but the evaluation only shows application scenarios and does not consider any performance/design discussion.

Reviewer 2 Report

In this paper, the authors propose an edge device framework for the IoT application server called SEMAR with the goal of remotely optimizing the edge device utilization. The topic considered by the authors is interesting although it only makes full sense in the context of the SEMAR system.

Therefore, the authors should add a "Generalization" section before the conclusions to clarify how many of the ideas proposed in this paper can be applied on platforms similar to SEMAR but different from it.

The authors detail the algorithms underlying their approach while the section on experiments is poor. The authors should add a section specifically designed to describe a test campaign whose goal is to demonstrate the effectiveness of the proposed approach. 

The Discussion section is extremely poor. Its content could be included in the new section on experiments. 

The authors should consider some more innovative IoT architectures proposed recently. Consider, for example, the SIoT (Social IoT - see the work of Iera et al.) architecture and the MIoT (Multi-IoT - see the works of Cauteruccio, Virgili et al.) architecture. In the new "Generalization" section I mentioned above, the authors should mention these architectures and indicate whether and to what extent their approach can be applied to them.

Finally, the authors should correct some typos in the paper (see, for instance, "SEMAR)" in the keywords